# Mesenchymal Stem Cells Restore Endothelial Integrity and Alleviate Emotional Impairments in a Diabetic Mouse Model via Inhibition of MMP-9 Activity

**DOI:** 10.3390/ijms26073355

**Published:** 2025-04-03

**Authors:** Aoying Chen, Yuhan Duan, Shaocong Zhou, Fangzhou Du, Huiyu Peng, Dongao Zeng, Jingwen Wang, Yue Wu, Shuaiguang Shi, Shikai Li, Shuang Yu, Jingzhong Zhang

**Affiliations:** 1School of Biomedical Engineering (Suzhou), Division of Life Sciences and Medicine, University of Science and Technology of China, Hefei 230026, China; cay0503@mail.ustc.edu.cn (A.C.); dyh0624@mail.ustc.edu.cn (Y.D.); zhousc97@mail.ustc.edu.cn (S.Z.); 15005215936@163.com (H.P.); zdabiosc@mail.ustc.edu.cn (D.Z.); sgshi@mail.ustc.edu.cn (S.S.); lishikai26@mail.ustc.edu.cn (S.L.); 2Suzhou Institute of Biomedical Engineering and Technology, Chinese Academy of Sciences, Suzhou 215163, China; du_fangzhou@163.com (F.D.); wangjw@sibet.ac.cn (J.W.); wuy@sibet.ac.cn (Y.W.); 3School of Medical Imaging, Xuzhou Medical University, Xuzhou 221004, China

**Keywords:** diabetes mellitus, emotional deficits, mesenchymal stem cells, blood–brain barrier, matrix metalloprotein-9

## Abstract

Diabetes mellitus (DM) has reached pandemic prevalence, significantly impacting global health. Accumulating evidence has highlighted a bidirectional relationship between diabetes and depression, with blood–brain barrier (BBB) disruption playing a pivotal role in the pathogenesis of and therapeutic approaches to both disorders. Mesenchymal stem cells (MSCs) have emerged as a promising cell-based therapeutic strategy for DM; however, their potential to mitigate DM-associated emotional deficits remains unclear. This study investigates whether MSCs can restore BBB integrity and improve emotional deficits in a diabetic mouse model via matrix metalloprotein-9 (MMP-9) inhibition. We used biochemical, molecular, and behavioral analyses to assess BBB function, inflammation, and emotional behavior. Our results demonstrated that diabetic conditions induce BBB dysfunction, characterized by the MMP-9-mediated degradation of tight junction (TJ) proteins claudin-5 (Cldn5) and occludin (Ocln), alongside neuroinflammation and emotional impairments. Notably, MSC administration restored BBB integrity and attenuated neuroinflammation by suppressing MMP-9 activity and upregulating TJ proteins. Importantly, MSC treatment not only alleviated anxiety- and depressive-like behaviors but also enhanced glycemic control in DMmodels. These findings elucidate the mechanistic basis of MSC therapy for DM-related neuropsychiatric complications and, crucially, reveal its dual therapeutic efficacy in concurrently ameliorating both neuropsychiatric symptoms and metabolic dysfunction in DM models. This synergistic therapeutic effect provides a translational rationale for advancing MSC-based therapies into clinical applications.

## 1. Introduction

Diabetes mellitus (DM), a global epidemic with rising prevalence, is increasingly recognized for its central nervous system (CNS) consequences [1,2,3]. While peripheral complications (e.g., retinopathy, nephropathy) are well characterized [4,5], emerging evidence reveals DM’s role in driving microvascular CNS pathologies (cognitive decline, stroke, Alzheimer’s disease) [6,7,8] and neuropsychiatric disorders [9,10]. Notably, recent studies have highlighted a significant bidirectional comorbidity between DM and depression [9,10], emphasizing the urgent need to unravel the shared pathophysiological mechanisms connecting these two conditions and to develop targeted therapeutic strategies.

The blood–brain barrier (BBB), a critical neurovascular interface maintaining CNS homeostasis [11], has emerged as a key mediator of diabetes-induced neurological complications. Fluctuations in plasma glucose levels induce structural and functional BBB alterations through multiple pathways, like oxidative stress in cerebral microcapillaries, the dysregulation of tight junction (TJ) proteins, and impaired transport mechanisms [12,13]. Of particular significance are TJ proteins, including claudin-5 (Cldn5) and occludin (Ocln), which form the structural backbone of BBB integrity [14]. Experimental evidence demonstrates that hyperglycemia and pro-inflammatory cytokines synergistically degrade these junctional complexes [15,16], particularly through the downregulation of Cldn5 (essential for barrier selectivity) and Ocln (critical for junction assembly) [17,18]. The resultant BBB hyperpermeability facilitates neuroinflammatory cascades [19,20], which are increasingly recognized as a common substrate linking DM with neuropsychiatric disorders.

Matrix metalloproteinase-9 (MMP-9), a zinc-dependent endopeptidase abundantly expressed in limbic regions [21], has recently been implicated in this pathological nexus. Functioning as a proteolytic enzyme, MMP-9 disrupts BBB integrity by degrading key structural components, including TJ proteins and extracellular matrix components, resulting in compromised barrier function. Elevated MMP-9 activity is observed in both DM and depression models [22,23,24]. This dysregulated proteolytic activity exacerbates BBB permeability, facilitating the infiltration of peripheral immune cells and inflammatory cytokines into the brain parenchyma [25,26,27]. Importantly, MMP-9 knockout models demonstrate preserved BBB integrity despite metabolic challenges [28], positioning this protease as a pivotal therapeutic target.

Current therapeutic options for DM patients with depression remain suboptimal. While antidiabetic medications demonstrate uncertain antidepressant efficacy, conventional antidepressants such as selective serotonin reuptake inhibitors (SSRIs) may worsen glycemic control [29,30]. Mesenchymal stem cells (MSCs) have emerged as a promising therapeutic strategy for DM and its complications due to their potent anti-inflammatory, immunomodulatory, and tissue-repairing properties [31]. Preclinical studies have demonstrated that MSCs can ameliorate hyperglycemia, improve insulin sensitivity [32,33], and alleviate DM complications, including neuropathy and nephropathy [34,35]. Moreover, MSCs have shown potential in mitigating depression-like behaviors and neuroinflammation in various animal models, possibly through their ability to modulate inflammatory pathways and promote tissue regeneration [36,37]. However, whether they can modulate MMP-9 activity, restore BBB integrity, and mitigate emotional deficits under DM conditions remains to be elucidated.

In this study, we utilized a high-fat diet/streptozotocin-induced (HFD/STZ) type 2 DM model to investigate the mechanisms underlying diabetes-associated neuropsychiatric complications and the therapeutic effects of MSCs. We hypothesize that MSCs restore BBB integrity and alleviate emotional deficits in diabetic mice by inhibiting MMP-9. Our results revealed that DM conditions lead to BBB dysfunction, characterized by the MMP-9-mediated degradation of TJ proteins, accompanied by neuroinflammation and emotional impairments. Through a comprehensive assessment of BBB integrity, neuroinflammatory markers, and depression-like behaviors, we demonstrated that MSC transplantation effectively restored BBB function and attenuated neuroinflammation by inhibiting MMP-9 activity and upregulating Cldn5 and Ocln expression. Importantly, MSC treatment not only alleviated emotional deficits but also improved glycemic control in DM models. These findings highlight the multifaceted therapeutic potential of MSCs in addressing diabetic neuropsychiatric complications and provide a mechanistic basis for their clinical translation.

## 2. Results

### 2.1. Diabetic Mice Exhibit Anxiety- and Depressive-like Behaviors

A diabetic mouse (DM) model was established using a high-fat diet (HFD) combined with streptozotocin (STZ) injections (Figure 1A). It is a well-established type 2 DM model recapitulating key pathophysiological features, including insulin resistance and pancreatic beta-cell dysfunction [38]. As shown in Figure 1B, DM mice displayed significantly higher blood glucose levels (>16.7 mM) compared to control (CTL) mice two weeks after STZ administration (Figure 1B). To assess glucose metabolism and insulin sensitivity, glucose tolerance tests (GTTs) and insulin tolerance tests (ITTs) were performed. In the GTT, blood glucose levels in DM mice were significantly elevated at all time points compared to CTL mice, with peak glucose levels observed at 30 min and maintained at elevated levels throughout the test. Additionally, the area under the curve (AUC) was significantly higher in DM mice than in CTL mice, indicating impaired glucose clearance (Figure 1C,D). Similarly, in the ITT, blood glucose levels in DM mice remained consistently higher at all time points compared to CTL mice, with a significantly larger AUC in the DM mice than in the CTL mice, confirming insulin resistance in the DM mice (Figure 1E,F). These results collectively demonstrate the successful establishment of the DM model.

Anxiety- and depressive-like behaviors are key indicators of emotional deficits associated with depression. In the open field test (OFT), the movement trajectory of DM mice was predominantly concentrated in the marginal areas of the arena (Figure 1G). Statistical analysis revealed that DM mice spent significantly less time (Figure 1H) and made fewer entries into the central zone (Figure 1I) compared to CTL mice. Consistent with these findings, in the elevated plus maze (EPM) test, the movement trajectory of DM mice was observed more on the closed arms (Figure 1J). Quantitative analysis showed that DM mice spent significantly less time in the open arms (Figure 1K). Both tests indicated that DM mice exhibited increased anxiety-like behaviors. Furthermore, in the tail suspension test (TST), DM mice demonstrated significantly longer immobilization time compared to CTL mice (Figure 1L), suggesting increased depressive-like behaviors in the DM group.

### 2.2. Increased Neuroinflammation in the Hippocampus of DM Mice

The hippocampus (HIP) is a key brain region within the emotional brain network, playing a crucial role in emotion processing in the brain. Moreover, synaptic deficits in HIP neurons lead to dysfunction in neural circuits that are essential for mood regulation [39]. Consistent with the increased anxiety- and depressive-like behaviors observed in DM mice, Western blot (WB) analysis revealed a significant reduction in the expression of the presynaptic protein synapsin I (SYNI) and the postsynaptic protein PSD95 in the HIP of DM mice compared to CTL mice (Figure 2A,B). These findings indicate significant synaptic deficits in the HIP of DM mice.

Microglia is pivotal in shaping synapses within the CNS during both development and adulthood [40]. In line with the observed synaptic deficits in the DM hippocampus, there was significant microglial activation, as evidenced by the increased number of Iba-1+ microglia in the HIP of DM mice (Figure 2C,D). Quantitative PCR (qPCR) analysis (Figure 2E) further demonstrated a significant increase in *iNOS*, a marker of pro-inflammatory M1 microglia, and a concomitant decrease in *Arg1*, a marker of anti-inflammatory M2 microglia, in the HIP of DM mice compared to CTL. This suggests that microglia are polarized toward an inflammatory phenotype in the DM state. Additionally, the expression levels of pro-inflammatory cytokines, including *IL-6*, *IL-1β*, and *TNF-α*, were significantly elevated in the HIP of DM mice compared to the CTL group (Figure 2F). Collectively, these findings indicate significant neuroinflammation in the HIP of DM mice, which likely contributes to the synaptic deficits associated with DM-related emotional disorders. Notably, the expression levels of pro-inflammatory cytokines, including IL-6, IL-1β, and TNF-α, were also significantly elevated in the peripheral blood of DM mice (Figure 2G), suggesting the presence of systemic inflammation in the DM state.

### 2.3. Impaired BBB Permeability and Enhanced MMP-9 Expression in the HIP of DM Mice

Given the presence of systemic and neuroinflammation in DM models, we investigated whether the BBB, a critical protective barrier in the brain, is compromised in DM. BBB permeability was assessed by introducing the low-molecular-weight fluorescent tracer sodium fluorescein (NaFI) into the peripheral circulation, followed by evaluating its penetration into brain tissue (Figure 3A). In contrast to CTL mice, DM mice displayed a notable accumulation of NaFI in the HIP (Figure 3B), indicating impaired BBB permeability in DM mice.

The BBB comprises microvascular endothelial cells (ECs) that line the cerebral capillaries, with endothelial TJ playing a pivotal role in maintaining BBB integrity [41]. WB analysis revealed a significant reduction in the expression levels of Cldn5 and Ocln, essential TJ components, in the HIP of DM mice compared to CTL (Figure 3C,D). Additionally, double immunostaining of Cldn5 and CD31, an EC marker, demonstrated discontinuous and fragmented Cldn5 localization within the CD31-labeled endothelium of DM mice (Figure 3E). Statistical analyses further indicated a significant decrease in the areas of Cldn5- and CD31-labeled TJs within CD31-labeled brain vascular endothelium (Figure 3F), suggesting impaired TJ protein expression and BBB integrity in the DM state.

Understanding the underlying mechanisms that disrupt the BBB would provide novel treatments for deficits linked to neuroinflammation. The activation of MMP-9 is known to compromise the BBB by breaking down TJ proteins such as Cldn5 and Ocln, consequently increasing BBB permeability in various pathological conditions [42]. In this study, WB analysis revealed a significant upregulation of MMP-9 at protein levels in the HIP of DM mice compared to CTL mice (Figure 4A,B). Consistent with these findings, elevated MMP-9 activity and a simultaneous reduction in TJ proteins Cldn5 and Ocln (Figure 4E,F) were observed in a brain vascular endothelial cell line bEND.3 when exposed to high glucose and saturated fatty acid palmitic acid (HG/HP), replicating hyperglycemia and metabolic dysfunction in diabetes.

Additionally, the impact of MMP-9 on TJ proteins was further investigated in bEND.3 cells. Treatment with the MMP-9 activator ginkgolide C significantly increased MMP expression and decreased Cldn5 and Ocln levels in bEND.3 cell cultures (Figure 4C,D). Conversely, pre-treatment with an MMP-9 inhibitor effectively attenuated the HG/HP-induced upregulation of MMP-9 and the subsequent reduction in Cldn5 and Ocln (Figure 4E,F), indicating that DM-related reductions in Cldn5 and Ocln are MMP-9-dependent. Notably, MMP-9 inhibition led to increased Cldn5 and Ocln expression, suggesting that targeting MMP-9 could serve as a potential strategy to enhance BBB integrity and function.

### 2.4. MSCs Suppress MMP-9 Expression and Restore BBB Integrity in the DM Hippocampus

As illustrated in Figure 5A, MSCs were administered intravenously at a dose of 2 × 10^6^ cells twice weekly. Therapeutic effects on BBB integrity, neuroinflammation, and the behavioral deficits were evaluated starting one week following the final injection.

Two-way ANOVA revealed significant interactions between DM and MSC treatment for MMP-9 expression, TJ protein levels, and BBB permeability, indicating that MSC treatment effectively rescued DM-induced deficits in these parameters. Specifically, WB analysis demonstrated that MSC administration significantly reduced DM-induced elevations in HIP MMP-9 protein levels (Figure 5B,C). Additionally, MSC treatment reversed the DM-associated reductions in TJ proteins, including Cldn5 and Ocln (Figure 5D,E). Immunofluorescence co-staining for Cldn5 and the endothelial marker CD31 further illustrated a significant increase in the ratio of Cldn5-stained TJs to CD31-stained endothelium in MSC-treated DM mice compared to untreated DM mice (Figure 5F,G), indicating a recovery of TJ density and structural complexity in DM mice following MSC transplantation. Moreover, the content of NaFI was significantly lower in the HIP of DM mice receiving MSC treatment during the BBB permeability assay, signifying improved BBB integrity in these animals (Figure 5H). Together, these results demonstrate that a two-week MSC regimen attenuates MMP-9 overactivity, enhances TJ protein expression, and rescues BBB integrity in the DM hippocampus.

### 2.5. MSCs Attenuate Neuroinflammation in the HIP of DM Mice

BBB disruption is a well-established driver of neuroinflammation and associated brain disorders [43]. Consistent with improved BBB integrity, MSC treatment significantly reduced the number of Iba-1+ microglia in the HIP of DM mice (Figure 6A,B). Additionally, qPCR analysis revealed a significant reduction in *iNOS* levels and a concurrent increase in *Arg1* expression in MSC-treated DM mice compared to untreated DM ones (Figure 6C). Analysis of pro-inflammatory cytokines, including *IL-6*, *IL-1β*, and *TNF-α*, revealed that MSC-treated DM mice exhibited significantly lower levels of these cytokines in the HIP compared to untreated DM mice (Figure 6D). These findings indicate that microglia are polarized toward an anti-inflammatory phenotype following MSC treatment. Two-way ANOVA demonstrated a significant interaction between DM and MSC treatment for microglial activation and inflammatory cytokine levels, indicating that MSCs effectively rescued DM-induced neuroinflammation. Notably, systemic inflammation, as evidenced by elevated IL-6, IL-1β, and TNF-α levels in the peripheral blood of DM mice, was also significantly reduced to control levels in the DM+MSC group (Figure 6E).

In parallel with reduced neuroinflammation, we observed elevated expression of synaptic markers, including presynaptic protein SYN1 and PSD95, in the HIP of MSC-treated DM mice compared to untreated DM mice (Figure 6F,G). Two-way ANOVA revealed a significant interaction between DM and MSC treatment for both synaptic proteins (*p* < 0.05), suggesting that MSC administration restored DM-induced synaptic deficits. Together, these findings demonstrate that MSCs effectively attenuate neuroinflammation in DM mice, thereby promoting synaptic remodeling in the HIP.

### 2.6. MSC Application Alleviated Emotional Deficits and Hyperglycemia in DM Mice

The OFT and EPM are used to evaluate the anxiety-like behaviors of mice. The OFT results, shown in Figure 7A–C, revealed that DM mice treated with MSCs exhibited a significant increase in the number of entries into the central area and the duration of time spent there compared to untreated DM mice. Similarly, the EPM results, presented in Figure 7D,E, demonstrated that DM mice treated with MSCs spent more time in the open arms. Together, these findings suggest that MSC treatment effectively alleviated anxiety-like behaviors in DM mice. Additionally, the TST was used to evaluate the depressive-like behaviors. The TST showed a reduction in immobility time in the DM+MSC group compared to the DM group, indicating a decrease in depressive-like behavior (Figure 7F). Taken together, these behavioral experiments provide strong evidence that MSC treatment can ameliorate both anxiety- and depression-like disorders in a DM mouse model.

The systemic metabolic effects of MSC administration were further investigated through longitudinal glycemic monitoring. While twice-weekly MSC injections over a 14-day period induced no significant alterations in body weight for either control or diabetic cohorts compared to their respective baseline groups (Figure 7G), marked improvements in glucose homeostasis were observed. DM mice receiving MSC treatment exhibited progressively reduced fasting blood glucose levels throughout the intervention period, achieving a 25% reduction by day 14 compared to untreated DM counterparts, but remained elevated relative to non-diabetic controls (Figure 7H). The GTT revealed sustained reductions in blood glucose concentrations across multiple time points (60, 90, and 120 min post-challenge) in MSC-treated DM mice (Figure 7I), with corresponding AUC values decreasing versus DM controls (Figure 7J). Similarly, the ITT demonstrated enhanced insulin responsiveness in the intervention group (Figure 7K), showing lower AUC compared to untreated DM animals (Figure 7L). Notably, while these metabolic parameters in MSC-treated DM mice remained statistically distinct from healthy controls, the observed attenuation of hyperglycemia and improved insulin sensitivity collectively demonstrate the therapeutic potential of MSC administration for modulating DM metabolic dysregulation.

## 3. Discussion

The present study collectively demonstrates that MSC therapy ameliorates diabetes-associated neuroinflammation and neuropsychiatric complications through a mechanism involving MMP-9 inhibition and BBB restoration. Crucially, these effects were paralleled by improved glycemic control and insulin sensitivity, suggesting that MSC therapy represents a potential novel strategy to achieve both metabolic stabilization and neurological recovery in DM. Unlike classical antidepressants that may worsen glycemic control [44], MSCs offer synergistic benefits by improving both CNS and peripheral metabolic parameters, thereby reducing polypharmacy risks in diabetic patients with neuropsychiatric complications. Moreover, MSCs target the shared pathophysiology of the DM and depression by repairing BBB leakage, resolving neuroimmune dysregulation, and normalizing metabolic parameters, ultimately improving outcomes and progressions for both conditions in patients.

The convergence of glycemic control and neuroinflammation reduction underscores the unique capacity of MSC to treat DM emotional deficits through multifaceted mechanisms. Consistent with previous reports, systemic improvements in hyperglycemia, insulin sensitivity, and reductions in peripheral inflammatory cytokines (e.g., TNF-α, IL-6) were observed in MSC-treated DM models [32,45]. These effects not only address the root cause of DM complications but also disrupt the vicious cycle of chronic inflammation and metabolic dysregulation, thereby alleviating peripheral drivers of neuropsychiatric dysfunction. Critically, the restoration of BBB integrity in the HIP, evidenced by upregulated TJ protein expression and suppressed MMP-9 activity, highlights MSCs’ central role in mitigating neuroinflammation. MMP-9, a key enzyme responsible for degrading TJ proteins and increasing BBB permeability [46], is significantly inhibited by MSC therapy, stabilizing endothelial junctions and preserving BBB integrity. The identified mechanisms underlying BBB disruption warrant further investigation to advance clinical translation. In particular, therapeutic targeting of endothelial TJ regulation may represent a promising strategy for neurological disorders characterized by BBB impairment, such as depression [47], Alzheimer’s disease [48], traumatic brain injuries [49], etc., where barrier dysfunction actively contributes to disease progression. Concurrently, the diminished activation of microglia and reduced expression of pro-inflammatory cytokines in the CNS confirm that MSC modulate neuroinflammation, potentially via the paracrine secretion of anti-inflammatory mediators or direct interactions with glial cells.

The reduction in neuroinflammation directly correlates with synaptic protein recovery in the HIP, a brain region pivotal to emotional regulation [50,51]. The HIP serves as a core hub within the emotional brain network, where the synaptic plasticity of HIP neurons plays a crucial role in the regulation of emotional behaviors [52,53]. Chronic neuroinflammation disrupts synaptic plasticity by downregulating key synaptic proteins, as demonstrated by their reduced levels in DM models. By suppressing inflammatory cascades and oxidative stress, MSCs foster a microenvironment conducive to synaptic remodeling in the HIP, which likely underpins the reversal of anxiolytic- and depressive-like behaviors observed in MSC-treated DM mice.

While this study provides critical mechanistic insights into MSC-mediated neuroprotection, several questions remain unresolved. First, the temporal and causal relationship between glycemic improvement and behavioral recovery requires further investigation. Specifically, the relative contribution of MSC-mediated hyperglycemia mitigation versus direct neuroinflammatory suppression to behavioral recovery remains unclear. Second, while the therapeutic effects of MSCs are attributed to their secretome, a spectrum of proteins, microRNAs, and extracellular vesicles, the identification of primary contributors is critical to developing more stable and efficient therapeutic strategies. Finally, the efficiency of intravenously administered MSCs to penetrate the BBB and injured brain regions remains inconsistent across preclinical studies. Addressing these mechanistic gaps will not only validate the therapeutic potential of MSCs but also refine their clinical application for DM and its neuropsychiatric complications. Future research should increase the targeting capacity of MSCs and optimize delivery routes to improve CNS biodistribution and the therapeutic efficacy of MSC therapy.

The clinical translation of MSC therapy faces challenges due to heterogeneity in MSC populations across donors and batches [54], which can significantly impact treatment outcomes. To address this, strict adherence to Good Manufacturing Practice (GMP) principles during MSC cultivation and preparation is essential for ensuring reproducible and reliable therapeutic effects. Recent advances in bioengineering techniques, including the use of precision-controlled bioreactors with microcarriers, are enabling large-scale, standardized MSC production, thereby minimizing batch-to-batch variability [55]. Furthermore, integrating advanced quality-control technologies such as Raman spectroscopy into the manufacturing workflow allows real-time monitoring of MSC product consistency [54], enhancing their clinical applicability.

In conclusion, our work positions BBB integrity as a potential therapeutic target in diabetic-associated emotional deficits and establishes MSC-based therapy as a promising strategy to concurrently address metabolic dysregulation, neuroinflammation, and emotional impairments. The capacity of MSCs to inhibit MMP-9 and enhance TJ protein expression provides a mechanistic foundation for clinical applications targeting diabetes-associated CNS disorders. Future research aimed at standardizing MSC production protocols and enhancing therapeutic efficacy for diabetes mellitus-associated depression will be critical to advancing the clinical translation of MSC-based therapies.

## 4. Materials and Methods

### 4.1. Preparation of MSC Culture

All studies involving human samples were conducted in compliance with the ethical guidelines for human embryonic stem cell research and the Declaration of Helsinki. Umbilical cords were obtained with signed informed consent and ethical approval from the First Affiliated Hospital of Soochow University (Approval No. 2019-136). MSCs were isolated and cultured following previously established protocols [56,57]. Briefly, the vessels and outer membrane of the umbilical cord were removed, and the mesenchymal tissue from Wharton’s jelly was dissected and minced into 1 mm^3^ pieces. These tissue blocks were then placed in 10 cm diameter culture dishes containing Dulbecco’s Modified Eagle’s Medium/Nutrient Mixture F-12 (DMEM/F12; Gibco, Carlsbad, CA, USA; C11330500BT) supplemented with 10% fetal bovine serum (FBS; Gibco; 10099-141) and 1% penicillin-streptomycin (P/S; Solarbio, Beijing, China; P1400). Half of the medium was replaced on the third day, and the entire medium was refreshed every three days afterwards. The tissue blocks were removed once cell confluence reached approximately 50%. MSCs at passages 4 to 6 (P4–P6) were selected for subsequent experiments. The characterization of MSCs has been performed and reported previously [57].

### 4.2. Endothelial Culture and Treatment

The bEnd.3 derived from mouse brain capillaries (Stem Cell Bank, Chinese Academy of Sciences; Shanghai, China, TCM40) were maintained in liquid nitrogen. To simulate the hyperglycemia and metabolic dysfunction associated with DM, bEnd.3 cells were exposed to 30 mM high glucose (HG) and 200 µM palmitic acid (PA) in DMEM culture medium for 24 h. Cells treated with DMEM culture medium were set as control (CON). In some experiments, bEnd.3 cells were pre-treated with 5 µM llomastat (GM6001; Selleck, Houston, TX, USA, S7157), a specific MMP-9 inhibitor (MMP9i), for 1 h prior to HG/PA treatment for an additional 24 h to block MMP-9 activity under HG/PA conditions. To further investigate the role of MMP-9 in endothelial protein regulation, bEnd.3 cells were treated with 10 µM ginkgolide C (Selleck, S3781), a specific MMP-9 activator (MMP9a), for 24 h.

### 4.3. Preparation of the DM Mouse Models

All animal experiments were conducted in accordance with the National Institutes of Health Guide for the Care and Use of Laboratory Animals and approved by the Biological Research Ethics Committee of the Chinese Academy of Sciences (Approval No. Ref.no. 2021–B13). The mice were maintained in a specific pathogen-free environment under a 12/12 h light–dark cycle. Adult male C57BL/6 J mice (7 weeks) (SPF Biotechnology, Beijing, China) were used as DM models as described previously [38]. Briefly, mice were fed a 60 kcal% fat diet (FBSH, Suzhou, China) for 1 week, followed by an intraperitoneal injection of streptozotocin (STZ; 50 mg/kg/day; Cayman, Ann Arbor, MI, USA; 13104) for 5 days. Blood glucose levels were monitored every 5 days after STZ injection from the tail vein using a glucometer (Accu-Chek Performa, Roche Diagnostics, Mumbai, MA, USA). Mice with fed blood glucose ≥16.7 mM (300 mg/L) for 2 weeks were considered as DM models [38]. Control mice were maintained on a standard diet. For the glucose tolerance test (GTT), mice were fasted for 12 h and then administered a single intraperitoneal injection of dextrose at a dose of 1.0 g/kg. For the insulin tolerance test (ITT), mice were fasted for 6 h followed by an intraperitoneal injection of insulin at a dose of 0.7 units/kg [58]. Blood samples were collected via tail nicks at predetermined time intervals.

### 4.4. Intravenous Injections of MSC

The control or DM mice were randomly (RAND; Microsoft Excel) assigned to either PBS or MSC treatment, resulting in four experimental groups: (1) CTL group: control mice receiving 0.2 mL of PBS; (2) MSC group: control mice receiving 2 × 10^6^ MSC in 0.2 mL of PBS; (3) DM group: DM mice receiving 0.2 mL of PBS; and (4) DM+MSC group: DM animals mice receiving 2 × 10^6^ MSC in 0.2 mL of PBS. All mice received intravenous injections via the tail vein once per week for a total of two injections. The therapeutic dose of 2 × 10^6^ MSCs was selected based on established protocols from prior preclinical studies [59,60]. For the tail vein injection, mice were restrained in a custom-designed Falcon tube with the tail protruding. The tail was warmed under a heating lamp to enhance vein visibility, and injections were administered into the lateral tail veins using a 1 mL syringe.

### 4.5. Behavioral Tests

At the end of the study (~16 weeks), mice underwent a series of behavioral tests to assess anxiety- and depressive-like behaviors. Anxiety-like behaviors were evaluated using the open field test (OFT) [61,62] and the elevated plus maze (EPM) test [63], while depressive-like behaviors were assessed using the tail suspension test (TST) [64]. All behavioral procedures were conducted during the animals’ light cycle. To minimize carry-over effects, the tests were performed with intervals of several days between each assessment. Video-recorded behaviors were analyzed using ANY-Maze software, version 7.16 (Stoelting, Kiel, WI, USA) in a blinded manner.

### 4.6. In Vivo BBB Permeability Assay

BBB permeability was assessed using sodium fluorescein dye (NaFI; Sigma, Tokyo, Japan, 518-47-8). Mice were injected intravenously in the tail vein with the fluorescent tracer 4% NaFI after 45 min to allow for systemic circulation of NaFI. Mice were then anesthetized with carbon dioxide and perfused with 0.9% saline to remove intravascular NaFI. The HIP were dissected, homogenized in 70% trichloroacetic acid (TCA), and centrifuged at 16,000× *g* for 20 min to remove precipitates. The fluorescence of the supernatant was measured at an excitation wavelength of 485 nm and an emission wavelength of 538 nm using a microplate reader (Thermo Scientific™ Varioskan™ LUX). NaFI uptake into the tissue was normalized to the corresponding blood fluorescence values.

### 4.7. Immunofluorescent Staining

Mice were euthanized with carbon dioxide, followed by perfusion with 4% PFA. Brains were carefully excised and sectioned at a thickness of 40 µm using a freezing microtome (Thermo Fisher Scientific, Waltham, MA, USA; Cryostar NX50). Brain sections were permeabilized with 0.15% Triton X-100, blocked with 5% fetal calf serum, and incubated overnight at 4 °C with the following primary antibodies: Cldn5 (Invitrogen, Waltham, MA, USA, 35-2500, 1:500) and CD31 (Invitrogen, ER-MP12, 1:1000); Iba1 (Wako, Monza, Italy, 019-19741, 1:1000). Immunoreactivity was visualized using appropriate Alexa Fluor-conjugated secondary antibodies and imaged with a confocal microscope (Nikon, Tokyo, Japan, A1R HD25). Cell nuclei were counterstained with DAPI (Beyotime, Shanghai, China; C1002) for 15 min. For each mouse, 5–7 brain slices were analyzed. Within each slice, five regions of interest (ROIs) were randomly selected, and microglial counts or the ratio of Cldn5+ area to CD31+ area within each ROI was quantified using ImageJ software, version 1.54. Data from all ROIs were averaged to generate final values per animal.

### 4.8. Enzyme-Linked Immunosorbent Assay (ELISA)

Serum samples obtained from blood collected via cardiac puncture prior to perfusion for immunostaining were analyzed for levels of IL-1β, IL-6, and TNF-α using corresponding mouse ELISA kits (SAB, College Park, MD, USA; IL-6, EK0499; IL-1β, EK0502; TNF-α, EK0497) according to the manufacturer’s instructions. Briefly, serum samples were added to pre-coated plates and incubated with specific primary antibodies. After washing, horseradish peroxidase (HRP)-conjugated secondary antibodies were added, and absorbance was measured at 450 nm using a microplate reader (Thermo Scientific™ Varioskan™ LUX). Cytokine concentrations were determined by comparing absorbance values to a standard curve.

### 4.9. Western Blot

Mice were euthanized with carbon dioxide, and the brains were quickly separated and rinsed in ice-cooled PBS. The hippocampi were dissected on an ice-cooled plate. The HIP tissues or cells were lysed in cell lysis buffer (Beyotime, Shanghai, China; P0013). After determining the protein concentration (Lowry method), samples were separated on 8% to 12% polyacrylamide gels and transferred to polyvinylidene fluoride (PVDF) membranes. The membranes were blocked and incubated overnight with the following antibodies: PSD95 (SYSN, Toronto, ON, Canada, 124002, 1:1000); SYN1 (SYSN, 106011, 1:1000); Cldn5 (Invitrogen, 35-2500, 1:1000); Ocln (Invitrogen, 71-1500, 1:1000); and MMP9 (Boster, Shanghai, China; PB9669, 1:1000). Specific protein bands were visualized using enhanced chemiluminescence (GE Life Sciences, Marlborough, MA, USA). After subtraction of the local background, the bands were quantified using ImageJ software, version 1.54.

### 4.10. Quantitative PCR (qPCR)

Total RNA from mouse brain samples or cells receiving various treatments was harvested with Trizol (Thermo, 15596018) and quantified using NanoDrop2000 (Thermo). qPCR was performed on a Bio-Rad CFX96 PCR System using TB Green Premix Taq (TAKARA, Kusatsu, Japan; RR042B) and appropriate primers [65]. The primers are described in Table 1. The housekeeping gene β-actin was used as an internal control.

### 4.11. Statistical Analysis

Numerical data are expressed as mean ± SEM. The data were subjected to unpaired *t*-tests or analysis of variance (ANOVA) followed by appropriate post hoc tests using GraphPad Prism version 9.0. The level of significance was set at *p* < 0.05.

## Figures and Tables

**Figure 1 ijms-26-03355-f001:**
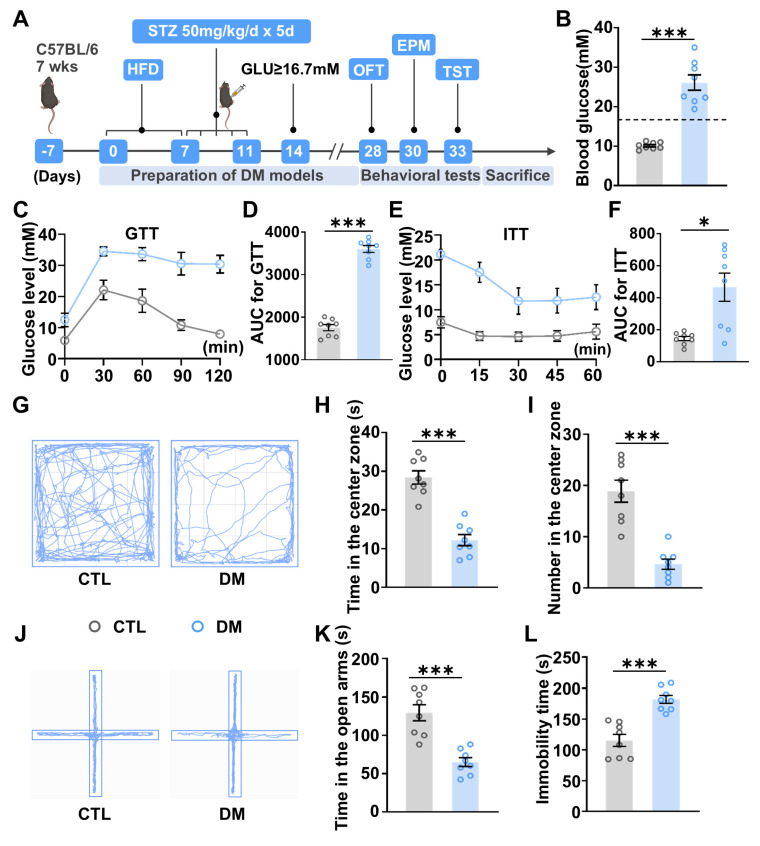
DM mice exhibit anxiety- and depressive-like behaviors. (**A**) Schematic diagram of the experimental design. Seven-week-old mice were fed a high-fat diet (HFD) for one week, followed by intraperitoneal injection of streptozotocin (STZ; 50 mg/kg/day, dissolved in citrate buffer, pH 4.5) for five consecutive days. Mice with blood glucose levels ≥16.7 mM (300 mg/dL) for two consecutive weeks were classified as DM and included in subsequent experiments. (**B**) Blood glucose levels in CTL and DM mice. The horizontal dotted line indicates a blood glucose of 16.7 mM. (**C**) Glucose tolerance test (GTT) results for CTL and DM mice. (**D**) Area under the curve (AUC) of GTT. (**E**) Insulin tolerance test (ITT) results for CTL and DM mice. (**F**) AUC of ITT. (**G**) Movement trajectories of CTL and DM mice in the OFT. The central zone is defined as the middle four cells of the arena. (**H**) Time spent in the central zone during the open field test (OFT). (**I**) Number of entries into the central zone during the OFT. (**J**) Movement trajectories of CTL and DM mice in the elevated plus maze (EPM) test, with open arms defined as the two non-walled opposing arms, the left and right arms are shown as open arms. (**K**) Time spent in the open arms during the EPM test. (**L**) Immobility time in the tail suspension test (TST). *n* = 8 for CTL and DM groups, respectively. Data are presented as mean ± SEM. Statistical significance was determined using a one-tailed unpaired *t*-test. * *p* < 0.05, *** *p* < 0.001.

**Figure 2 ijms-26-03355-f002:**
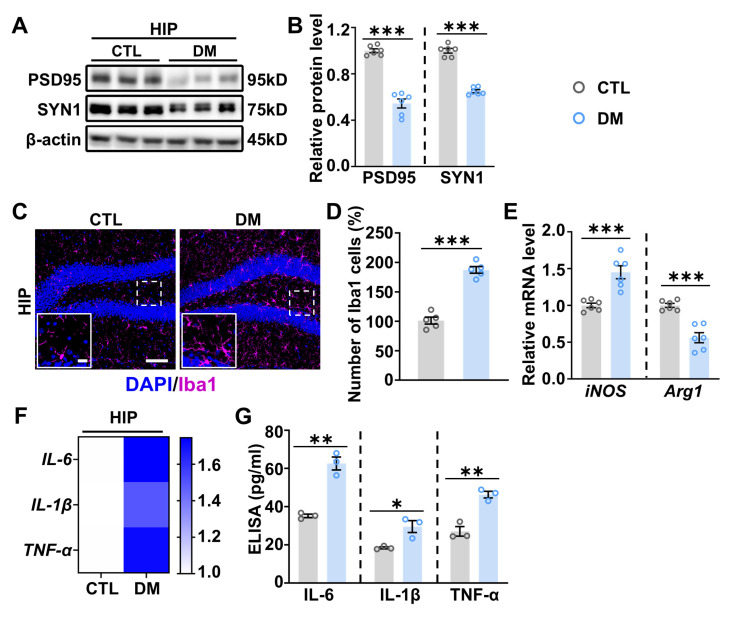
DM mice exhibit synaptic deficits and increased neuroinflammation in the HIP. (**A**) Representative WB images showing the expression levels of the presynaptic protein synapsin I (SYN1) and the postsynaptic protein postsynaptic density protein 95 (PSD95) in the HIP of CTL and DM mice. β-actin was used as an internal control. (**B**) Semi-quantitative analysis of PSD95 and SYN1 expression levels from immunoblot experiments. *n* = 6. (**C**) Representative images of Iba1 immunostaining in the HIP of CTL and DM mice. The area within the dashed box is magnified and displayed in the insets. Scale bars: 100 µm (main images) and 20 µm (insets). (**D**) Quantitative analysis of Iba1-positive microglia in the HIP of CTL and DM mice. *n* = 5. (**E**) qPCR analysis of *iNOS* and *Arg1* mRNA levels in the HIP of CTL and DM mice. *n* = 6. (**F**) qPCR analysis of *IL-6*, *IL-1β*, and *TNF-α* mRNA levels in the HIP of CTL and DM mice. *n* = 6. (**G**) ELISA measuring IL-6, IL-1β, and TNF-α levels in the serum of CTL and DM mice. *n* = 3. Data are presented as mean ± SEM. Statistical significance was determined using a one-tailed unpaired *t*-test. * *p* < 0.05, ** *p* < 0.01, *** *p* < 0.001.

**Figure 3 ijms-26-03355-f003:**
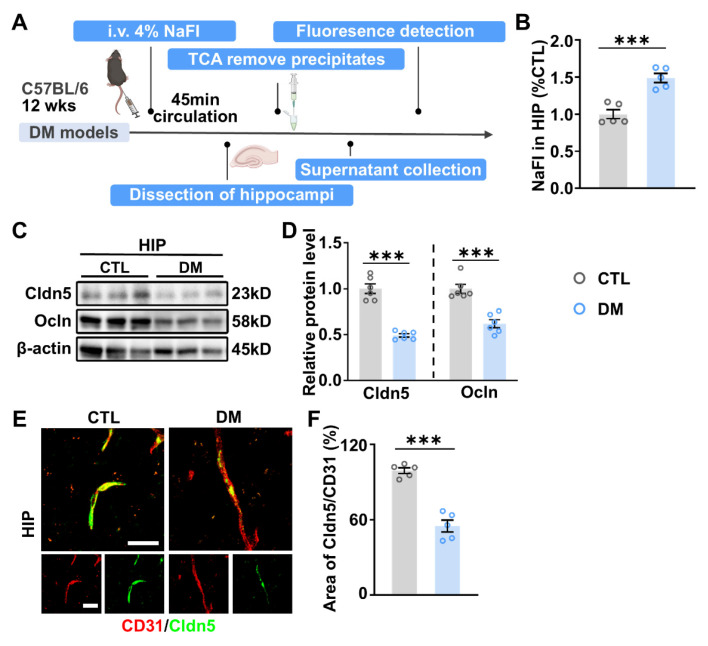
Impairments in BBB permeability and TJ protein expression in DM mice. (**A**) Schematic illustration of the BBB permeability assay utilizing the fluorescent tracer sodium fluorescein (NaFI). (**B**) Relative levels of NaFI in the HIP of CTL and DM mice. *n* = 5. (**C**) Representative WB images depicting the expression levels of claudin-5 (Cldn5) and occludin (Ocln) in the HIP of CTL and DM mice. β-actin was employed as an internal control. (**D**) Semi-quantitative results of Cldn5 and Ocln immunoblot analysis. *n* = 6. (**E**) Representative images displaying the double immunostaining of CD31 and Cldn5 in the HIP of CTL and DM mice. The merged stainings are displayed in the upper panel, while the staining of the individual channels is shown in the lower panel. Scale bars: both 10 µm in main images and insets. (**F**) Quantitative analysis of the ratios of Cldn5 and CD31-labeled areas to the CD31-labeled areas. *n* = 5. Data are presented as mean ± SEM. Statistical significance was evaluated using a one-tailed unpaired *t*-test. *** *p* < 0.001.

**Figure 4 ijms-26-03355-f004:**
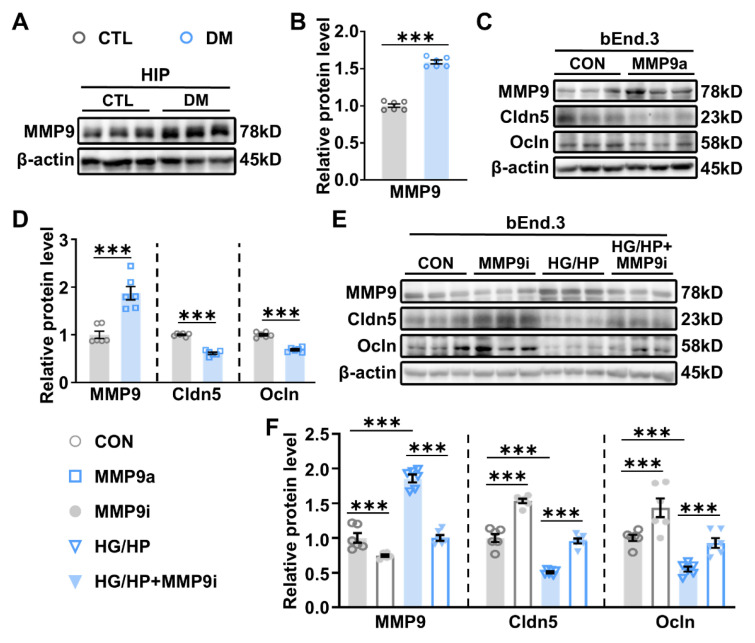
MMP-9 activation is involved in the reduced expression of TJ proteins. (**A**) Representative WB images showing the expression levels of MMP-9 in the HIP of CTL and DM mice. β-actin was used as an internal control. (**B**) Semi-quantitative results of MMP-9 immunoblot analysis. *n* = 6. (**C**) Representative WB images showing the expression levels of MMP-9, Cldn5, and Ocln in bEnd.3 cells after treated with 10 µM ginkgolide C (Selleck, Houston, USA, S3781), a specific MMP-9 activator (MMP9a), for 24 h. β-actin was used as an internal control. (**D**) Semi-quantitative results of MMP-9, Cldn5, and Ocln immunoblot analysis. *n* = 6. (**E**) Representative WB images showing the expression levels of MMP-9, Cldn5, and Ocln in bEnd.3 cell treated with 5 µM llomastat, a specific MMP-9 inhibitor (MMP9i), for 1 h prior to high glucose and saturated fatty acid palmitic acid (HG/HP) treatment for an additional 24 h. A total of 30 mM high glucose and 200 µM palmitic acid (PA) (HG/HP) were used to mimic the hyperglycemia and metabolic dysfunction in DM state. β-actin was used as an internal control. (**F**) Semi-quantitative results of MMP-9, Cldn5, and Ocln immunoblot analysis. *n* = 6. Data are presented as mean ± SEM. One-tailed unpaired *t*-test for (**B**,**D**) and two-way ANOVA followed by Tukey’s post hoc test for (**F**). *** *p* < 0.001.

**Figure 5 ijms-26-03355-f005:**
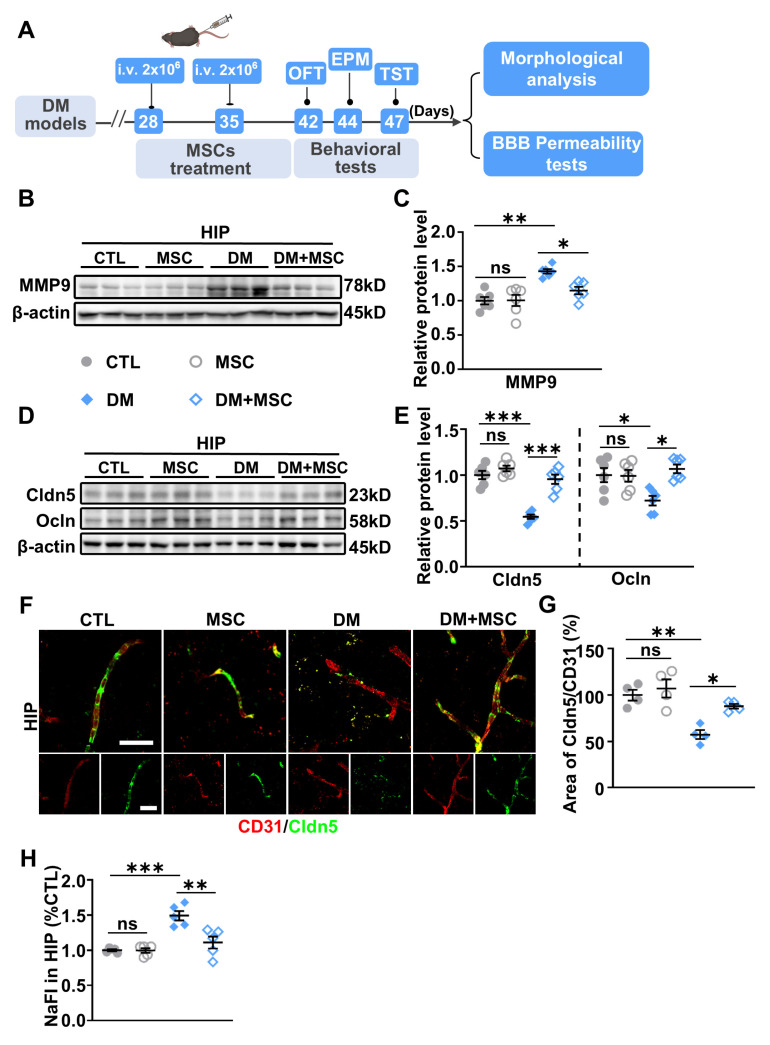
MSC treatment suppresses MMP-9 expression and restores BBB integrity. (**A**) Schematic of the experimental design for MSC administration in DM mouse models. After DM mice were established (~12 weeks), 2 × 10^6^ MSCs in 0.2 mL PBS were injected intravenously via tail vein once weekly for two weeks. CTL mice received PBS injections and were maintained on a normal diet. Mice were assessed for behavioral tests (OFT, EPM, TST), and subsequently sacrificed for molecular analyses, morphological analyses, and BBB permeability. (**B**) Representative WB images of MMP-9 protein levels in the HIP of CTL, MSC, DM, and DM+MSC groups. β-actin served as the loading control. (**C**) Quantified densitometric analysis of MMP-9 protein expression. *n* = 6. (**D**) Representative WB images of tight junction (TJ) proteins Cldn5 and Ocln in the HIP of CTL, MSC, DM, and DM+MSC mice. β-actin was used as the loading control. (**E**) Quantified densitometric analysis of Cldn5 and Ocln protein levels. *n* = 6. (**F**) Representative immunofluorescence images of double staining for CD31 and Cldn5 in the HIP of CTL, MSC, DM, and DM+MSC mice. Scale bars: both 10 µm in main images and insets. (**G**) Quantification of TJ density, represented by the ratio of Cldn5+CD31+ areas to CD31+ endothelial areas. *n* = 4. (**H**) Relative accumulation of NaFI in the HIP, indicating BBB permeability. *n* = 5. Data are presented as mean ± SEM. Two-way ANOVA followed by Tukey’s post hoc test. * *p* < 0.05, ** *p* < 0.01, *** *p* < 0.001.

**Figure 6 ijms-26-03355-f006:**
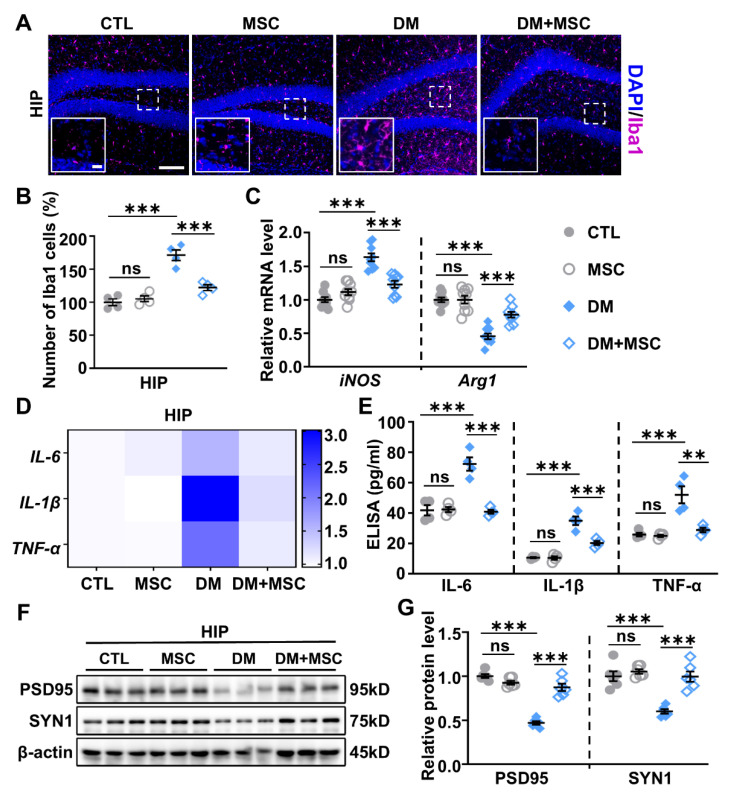
MSC treatment attenuated DM-induced neuroinflammation and synaptic deficits. (**A**) Representative immunofluorescence images of Iba1 staining in the HIP of mice from the CTL, MSC, DM, and DM+MSC groups. The dashed box indicates the magnified region shown in the insets. Scale bars: 100 µm (main images) and 20 µm (insets). (**B**) Quantitative analysis of Iba1+ microglia in the HIP. *n* = 4. (**C**) qPCR analysis of *iNOS* and *Arg1* mRNA levels in the HIP of mice from the CTL, MSC, DM, and DM+MSC groups. *n* = 9. (**D**) qPCR analysis of mRNA levels of *IL-6*, *IL-1β*, and *TNF-α* in the HIP of mice from the CTL, MSC, DM, and DM+MSC groups. *n* = 9. (**E**) ELISA results showing serum levels of IL-6, IL-1β, and TNF-α in CTL, MSC, DM, and DM+MSC mice. *n* = 4. (**F**) Representative WB images of PSD95 and SYN1 expression in the HIP. β-actin was used as a loading control. (**G**) Semi-quantitative analysis of PSD95 and SYN1 protein levels. *n* = 6. Data are presented as mean ± SEM. Two-way ANOVA followed by Tukey’s post hoc test. ** *p* < 0.01, *** *p* < 0.001.

**Figure 7 ijms-26-03355-f007:**
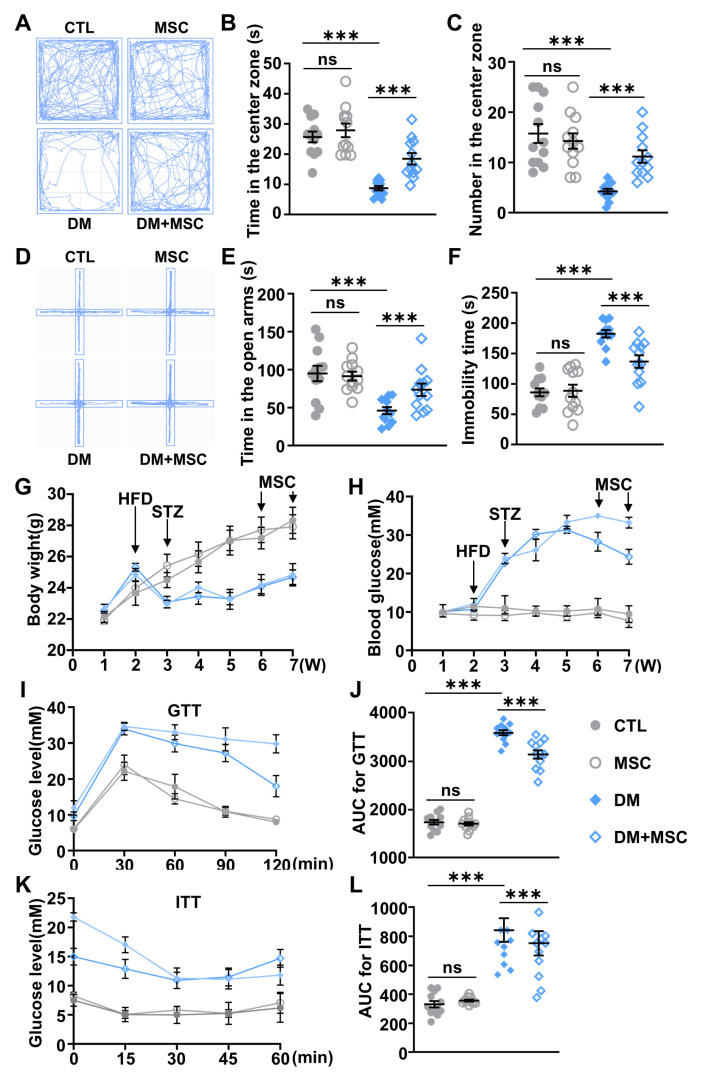
MSC treatment attenuates neuropsychiatric and metabolic dysregulation in DM mice. (**A**) Representative motion trajectory of mice in CTL, DM, MSC, and DM+MSC groups in the OFT. The central zone is defined as the middle four cells. (**B**) Quantification of time spent in the central zone during the open field test (OFT), reflecting anxiety-like behavior. (**C**) Frequency of entries into the central zone. (**D**) Motion traces of mice in the elevated plus maze (EPM), with open arms defined as the two non-walled opposing arms, the left and right arms are shown as open arms. (**E**) Percentage of time spent in open arms during EPM, indicative of anxiolytic effects. (**F**) Immobility duration in the tail suspension test (TST), quantifying depression-like behavioral despair. (**G**) Body weight dynamics during DM model development and subsequent MSC intervention, plotted across the 7-week experimental timeline. (**H**) Weekly fasting blood glucose measurements tracking metabolic progression from baseline throughout the study duration. (**I**) Glucose tolerance test (GTT) curves following intraperitoneal glucose challenge. (**J**) Area under the curve (AUC) analysis of GTT data, normalized to baseline glucose levels. (**K**) Insulin tolerance test (ITT) curves after insulin injection. (**L**) AUC quantification of ITT responses, reflecting systemic insulin sensitivity. *n* = 12. Data are presented as mean ± SEM. One-tailed unpaired *t*-test for (**J**,**L**); and two-way ANOVA followed by Tukey’s post hoc test for (**B**,**C**,**E**,**F**). *** *p* < 0.001.

**Table 1 ijms-26-03355-t001:** Primer sequences for qPCR.

Gene Names	Primer Sequences (5′-3′)
*iNOS*	forward: TTCACCCAGTTGTGCATCGACCTA
	reverse: AACTCCAATCTCGGTGCCCATGTA
*Arg1*	forward: TTGGCAAGGTGATGGAAGAGACCT
	reverse: CGAAGCAAGCCAAGGTTAAAGCCA
*IL-6*	forward: TAGTCCTTCCTACCCCAATTTCC
	reverse: TTGGTCCTTAGCCACTCCTTC
*IL-1β*	forward: GCAACTGTTCCTGAACTCAACT
	reverse: ATCTTTTGGGGTCCGTCAACT
*TNF-α*	forward: GCCACCACGCTCTTCTGTCT
	reverse: TGAGGGTCTGGGCCATAGAAC
*β-actin*	forward: TGCTGACAGAGGCACCACTGAA
	reverse: CAGTTGTACGTCCAGAGGCATAG

## Data Availability

The data that support the findings of this study are available from the corresponding author upon reasonable request.

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
