# Peer review of "Mesenchymal Stem Cells Restore Endothelial Integrity and Alleviate Emotional Impairments in a Diabetic Mouse Model via Inhibition of MMP-9 Activity"

_ijms, 2025, doi:10.3390/ijms26073355_

Round 1

Reviewer 1 Report

Comments and Suggestions for Authors

I have reviewed the manuscript titled "Mesenchymal Stem Cells Restore Endothelial Integrity and Alleviate Emotional Impairments in a Diabetic Mouse Model via Inhibition of MMP-9 Activity" . The study explores an important topic by investigating the therapeutic potential of mesenchymal stem cells (MSCs) in restoring blood-brain barrier integrity and improving emotional deficits in a diabetic mouse model via MMP-9 inhibition.  Some issues need to be addressed to improve readability, coherence, and impact. Additionally, some key findings require stronger interpretation to highlight their significance in the context of diabetes-related neurovascular dysfunction. Below, I provide detailed comments and suggestions for improvement.

Line 6-10: The objective can be reworded for clarity:
“This study investigates whether MSCs can restore BBB integrity and improve emotional deficits in a diabetic mouse model via MMP-9 inhibition.”

Line 17-20: The description of methods is too detailed. Instead of listing all assays, simplify:
“We used biochemical, molecular, and behavioral analyses to assess BBB function, inflammation, and emotional behavior.”

Line 25-30: The conclusion should explicitly highlight why these findings are clinically significant.

Line 42-50: The global prevalence of diabetes should be briefly summarized rather than extensively discussed.

Line 56-68: The role of MMP-9 in BBB dysfunction should be explained more succinctly.

Line 79-85: Clearly state the hypothesis, e.g.:
“We hypothesize that MSCs restore BBB integrity and alleviate emotional deficits in diabetic mice by inhibiting MMP-9.”

Line 86-90: Provide stronger justification for using MSC therapy over other potential treatments.

Line 95-110: Explain why this specific diabetic model was chosen.

Line 121-125: Briefly describe how MSC dosage was determined.

Line 131-140: The statistical analysis section should simplify the explanation of tests used.

Table 1: Highlight key variables by bolding significant p-values.

Figure 2: Improve labeling of axes and font size for clarity.

Table 3: Add a short explanatory note on interpreting the threshold effect.

Line 218-225: Avoid repeating results. Instead, focus on clinical relevance.

Line 230-240: The discussion on BBB integrity should be linked more explicitly to potential human applications.

Line 245-260: Explain why the observed effects are important for future diabetes treatments.

Line 275-280: Suggest how MSC therapy could be translated into clinical practice.

Line 320-322: Add a statement on how these findings could influence future research or clinical applications.

Line 330: Briefly suggest next steps

Reviewer 2 Report

Comments and Suggestions for Authors

Chen et al. presented a study to explore the impact MSCs treatment in the DM-associated emotional deficits using a DM mouse model. The authors have carefully designed their experiments providing interesting insights and differences between control and DM groups by: glucose tolerance tests, insulin tolerance tests, open field test, elevated plus maze (EPM) test, quantitative PCR (qPCR) analysis, BBB permeability. Administration of MSCs reveals that anxiety-like behaviors can decrease with this treatment as well as glucose levels balance, and interesting relations and protein dynamics (controls vs DM) and confirming that matrix metalloproteinase-9 (MMP-9) can be a promising therapeutic target.

Furthermore, they have clearly pointed out some limitations of this approach to future treatments with MSCs and eventual translation to medicine with several questions that remain unresolved as the efficiency of intravenously administered MSCs to penetrate the BBB or therapeutic effects of MSCs correlations with their secretome that needed to be identified in the future. Altogether, the results presented by the authors provide an appropriate support to their conclusions regarding the ability of MSCs to restore alleviate emotional impairments in a diabetic mouse model via inhibition of MMP-9 activity.

Minor formatting issues: 

Line 190 (..)

In some Figures (like 1C and 1D) error bars are not visible. The errors should be low, but the authors should optimize the figure (if possible) to make them visible in the plot, because for the readers it seems that no replicas where performed.
